# A Unified Causal View of Domain Invariant Supervised Representation Learning

## Abstract

Machine learning methods can be unreliable when deployed in domains that differ from the domains on which they were trained. One intuitive approach for addressing this is to learn representations of data that are domain-invariant in the sense that they preserve data structure that is stable across domains, but throw out spuriously-varying parts. There are many approaches aimed at this kind of representation-learning, including methods based on data augmentation, distributional invariances, and risk invariance. Unfortunately, it is often unclear when a given method actually learns domain-invariant structure, and whether learning domain-invariant structure actually yields robust models. The key issue is that, in general, it's unclear how to formalize "domain-invariant structure". The purpose of this paper is to study these questions in the context of a particular natural domain shift notion that admits a natural formal notion of domain invariance. This notion is a formalization of the idea that causal relationships are invariant, but non-causal relationships (e.g., due to confounding) may vary. We find that whether a given method learns domain-invariant structure, and whether this leads to robust prediction, both depend critically on the true underlying causal structure of the data.

## 1 Introduction

Machine learning methods could have unreliable performance at the presence of *domain shift* (Shimodaira, 2000; Quinonero-Candela et al., 2008), a structural mismatch between the training domain(s) and the deployed domain(s). A variety of techniques have been proposed to mitigate domain shift problems. One popular class of approach—which we'll focus on in this paper—is to try to learn a representation function $\phi$ of the data that is in some sense "invariant" across domains. Informally, the aim of such methods is to find a representation that captures the part of the data structure that is the "same" in all domains while discarding the part that varies across domains. It then seems intuitive that a predictor trained on top of such a representation would have stable performance even in new domains. Despite the intuitive appeal, there are fundamental open questions: when does a given method actually succeed at learning the part of the data that is invariant across domains? When does learning a domain invariant representation actually lead to robust out-of-domain predictions?

There are many methods aimed at domain-invariant representation learning. When applied to broad ranges of real-world domain-shift benchmarks, there is no single dominant approach, and an attack that works well in one context is often worse than vanilla empirical risk minimization in another (i.e., just ignoring the domain-shift problem). We'll study three broad classes of method:

**Data augmentation** Each example is perturbed in some way and we learn a representation that is the same for all perturbed versions. E.g., if $t(X)$ is a small rotation of an image $X$, then $\phi(X) = \phi(t(X))$ (Krizhevsky et al., 2012; Hendrycks et al., 2019; Cubuk et al., 2019; Xie et al., 2020; Wei & Zou, 2019; Paschali et al., 2019; Hariharan & Girshick, 2017; Sennrich et al., 2015; Kobayashi, 2018; Nie et al., 2020).

**Distributional invariance** We learn a representation so that some distribution involving $\phi(X)$ is constant in all domains. There are three such distributional invariances that can be required to hold for all domains $e, e'$:

    **marginal invariance**: $\mathrm{P}^e(\phi(X)) = \mathrm{P}^{e'}(\phi(X))$ (Muandet et al., 2013; Ganin et al., 2016; Albuquerque et al., 2020; Li et al., 2018a; Sun et al., 2017; Sun & Saenko, 2016; Matsuura

     & Harada, 2020);

     **conditional invariance**: When $Y$ is a label of interest, $\mathrm{P}^e(\phi(X) \mid Y) = \mathrm{P}^{e'}(\phi(X) \mid Y)$ (Li et al., 2018b; Long et al., 2018; Tachet des Combes et al., 2020; Goel et al., 2020)

     **sufficiency**: $\mathrm{P}^e(Y \mid \phi(X)) = \mathrm{P}^{e'}(Y \mid \phi(X))$ (Peters et al., 2016; Rojas-Carulla et al., 2018; Wald et al., 2021).

**Risk minimizer invariance**  For supervised learning, we learn a representation $\phi(X)$ so that there is a fixed (domain-independent) predictor $w^*$ on top of $\phi(X)$ that minimizes risk in all domains (Arjovsky et al., 2019; Lu et al., 2021; Ahuja et al., 2020; Krueger et al., 2021; Bae et al., 2021).

In each case, the aim is to learn a representation that throws away information that varies 'spuriously' across domains while preserving information that is reliably useful for downstream tasks. However, the notion of what is thrown away is substantively different across all the approaches and it is unclear which, if any, is appropriate for any particular problem.

The principle challenge to answering our motivating questions is that it's unclear in general how to formalize the idea of "part of the data that is invariant across domains". To make progress, it is necessary to specify the manner in which different domains are related. In particular, we require an assumption that is both reasonable for real-world domain shifts and that precisely specifies what structure is invariant across domains. In many problems, it is natural to assume that causal relationships—determined by the unobserved real-world dynamics underpinning the data—should be the same in all domains. We'll use an assumption of this form; the first task is to translate it into a concrete notion of domain shift.[1]

Specializing to supervised learning with label $Y$ and covariates $X$, we proceed as follows. The covariates $X$ are caused by some (unknown) factors of variation. These factors of variation are also dependent with $Y$. For some factors of variation, jointly denoted as $Z$, the relationship between $Y$ and $Z$ is spurious in the sense that $Y$ and $Z$ are dependent due to an unobserved common cause. The distribution of this unobserved common cause may change across environments, which in turns means the relationship between $Y$ and $Z$ can shift. However, the structural causal relationships between variables will be the same in all environments—e.g., $\mathrm{P}(X \mid \mathrm{pa}(X))$ is invariant, where $\mathrm{pa}(X)$ denotes the (possibly unobserved) causal parents of $X$. We call a family of domains with this structure *Causally Invariant with Spurious Associations* (CISA).

Concretely, consider the problem of classifying images $X$ as either Camel or Cow $Y$. In training, the presence of sand in the image background $Z$ is strongly associated with camels. But, we may deploy our new classifier in an island domain where cows are frequently on beaches—changing the $Z$-$Y$ association. Nevertheless, the causal relationships between the factors of variation—$Y$, $Z$, and others such as camera type or time of day—and the image $X$ remain invariant.

In this example, a natural formalization of "domain invariant part" of the image is the part that does not changes if grass $Z$ is added or removed from the background; an invariant representation learning method should learn a $\phi(X)$ that throws away such information. The aim of CISA is to be a reasonably broad notion of domain-shift that also allows us to formalize this intuitive notion of domain-invariance. Namely, under CISA, we define the domain-invariant part of $X$ to be the part that is not causally affected by the spurious factors of variation $Z$. Accordingly, a representation $\phi$ is domain invariant if $\phi(X)$ is not causally affected by $Z$. We say a representation with this property is *counterfactually invariant to spurious factors* (CF-invariant for short).

We now return to the motivating questions: when does a given method actually succeed at learning the part of the data that is invariant across domains? And, when does learning a domain invariant representation actually lead to more robust out-of-domain predictions? In the context of CISA-compatible domain shifts, we can answer the first question by determining the conditions under which each approach learns a CF-invariant representation, and the second question by studying the relationship between CF-invariance and domain shift performance.

Informally, the technical contributions of the paper are:

    1. Formalization of CISA and Counterfactually Invariant Representation Learning.

---

[1]This kind of causal-invariance assumption is already used in the domain-shift literature, though the formal domain shift notion we'll use here differs from previous approaches, see Section 2.

2. A characterization of the causal (real-world) conditions where each domain invariant representation learning method yields CF-invariant representations. This turns out to yield both a hierarchy of methods, and guidance for which methods are appropriate for a given problem.

3. Results relating invariant learning to robust learning. We'll find that, in general, there is not a simple relationship. For instance, learning a representation via risk minimizer invariance (an intuitive formalization of robustness) is impossible in general. However, we'll also find that there are (common) cases where there is a close connection. These results highlight the fundamental role of the underlying causal structure of the data.

## 2 RELATED WORK

**Causal invariance in domain generalization** Several works (Peters et al., 2016; Rojas-Carulla et al., 2018; Arjovsky et al., 2019; Lu et al., 2021) specify domain shifts with causal models. In these frameworks, the invariant predictor learns $P(Y \mid \mathrm{pa}(Y))$. However, this rules out, e.g., cases where $Y$ causes $X$ (Schölkopf et al., 2012; Lipton et al., 2018). CISA allows learning with more general structures, which we'll see is helpful in understanding the representation learning methods.

**Domain generalization methods** There are many methods for domain generalization; we give a categorization in the introduction. There have been a number of empirically-oriented surveys testing domain generalization methods in natural settings (Koh et al., 2021; Wiles et al., 2021; Gulrajani & Lopez-Paz, 2020). These find that no method consistently beats ERM, but many methods work well in at least some situations. Our aim here is to give theoretical insight into when each might work.

**Robust methods** The CISA framework is reasonable for many real-world problems, but certainly not all. There are other notions of domain shifts and differently motivated methods that do not fit under this framework. For example, many works (e.g., Sagawa et al., 2019; Liu et al., 2021; Ben-David et al., 2022; Eastwood et al., 2022) assume the testing domains are not too different from the training domains (e.g., test samples are drawn from the mixture of training distributions).These methods are complementary to the invariant representation learning approaches we study here; e.g., they would all fail in the two-bit-environments experiment in Appendix A.

## 3 CAUSAL INVARIANCE WITH SPURIOUS ASSOCIATIONS

**Setup** In domain generalization problems, we have training and test data from several related domains. The goal is to learn a predictor using data from training domains, and apply it to unseen test domains. The data comes in the form of $(X, Y, E)$ where $X \in \mathcal{X}$ is the input, $Y \in \mathcal{Y}$ the label and $E \in \mathcal{E}$ is the domain index. We abstract each domain as a probability distribution $P_e$, where $X_i, Y_i \mid E_i = e \overset{\text{iid}}{\sim} P_e$. At training time, we have access to data from a finite set of domains $\mathcal{E}_{\text{train}} \subset \mathcal{E}$.

Our goal is to learn a representation $\phi$ so that $\phi(X)$ is in some sense invariant across the domains. Many methods (e.g., Arjovsky et al., 2019; Ganin et al., 2016; Goel et al., 2020) frame their learning procedures this way, though there are many distinct formal notions of "invariance". The first task is to establish a canonical notion of invariance.

**Causal Invariance with Spurious Associations** All methods for handling domain shifts must specify the structure that is preserved across domains, and the ways in which they are allowed to vary.[2] Here, we'll rely on a particular variant of the assumption that causal relationships are held fixed, while non-causal relationships may vary. To formalize this, we assume that $X$ is caused by some unobserved factors of variation, and give a notion of what it means for these factors to be spuriously associated with $Y$:

**Definition 1.** We say a latent cause of $X$ is a *spurious factor of variation* if it is not a cause of $Y$ and there is some latent confounder that affects (directly or indirectly) both it and $Y$. Call the set of all such causes the *spurious factors of variations*.

Figure 1 shows examples of causal structures for prediction problems involving spurious factors of variation ($Z$). $X$ is divided into two parts: $X_{\bar{z}}^{\perp}$ denotes the part of $X$ not affected by the spurious

---

[2]Without such an assumption, the test domain could be chosen adversarially to have support only on points where the training-domain predictor makes mistakes.

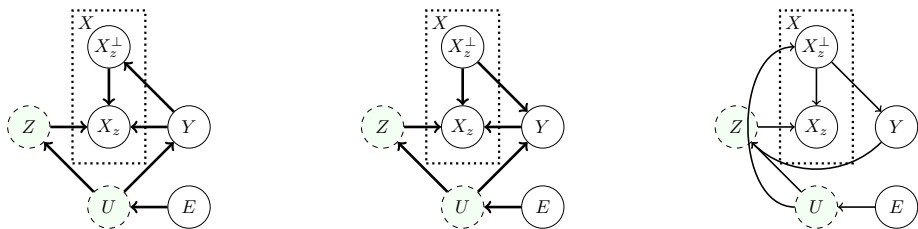

**Figure 1:** Examples of CISA-compatible causal structures. In each case, all causal relationships (solid arrows) are invariant across domains. Only the distribution of the unobserved confounder $U$ can vary, changing the induced association between $Y$ and the unobserved spurious factors $Z$.

factors $Z$, and $X_z$ the remaining part affected by $Z$. We do not assume the division of $X$ is known a priori (indeed, $Z$ is not even known!). Also note that $X_z^\perp$ may affect $X_z$, so these parts need not be independent. Formally, we use the notion of counterfactual invariance from Veitch et al. (2021).

**Definition 2.** Let $\phi$ be a function from $\mathcal{X}$ to $\mathcal{H}$ and $Z$ the spurious factors. We say $\phi$ is *counterfactually invariant to spurious factors* (abbreviated CF-invariant), if $\phi(X(z)) = \phi(X(z'))$ *a.e.*, $\forall z, z' \in \mathcal{Z}$. Here, $X(z)$ is potential outcomes notation, denoting the $X$ we would have seen had $Z$ been set to $z$.

Then, we can formalize the part of $X$ that's not affected by $Z$:

**Definition 3.** $X_z^\perp$ is a $X$-measurable variable such that, for all measurable functions $f$, $f$ is CF-invariant iff $f(X)$ is $X_z^\perp$-measurable.[3]

Now, we characterize how the relationship between $Z$ and $Y$ changes across domains with a definition of *unobserved confounder*.

**Definition 4.** We say an unobserved variable is an *unobserved confounder* if it is a common cause of $Z$ and $Y$, does not confound the relationship between $X_z^\perp$ and $Y$, and is caused by $E$ alone. Call the set of all such confounders *unobserved confounders*.

We introduce unobserved confounders $U$ that induces the spurious correlations between $X_z$ and $Y$. Now put together all the ingredients and formally specify how the domains are related:

**Definition 5.** We say a set of domains $\mathcal{E}$ are *Causally Invariant with Spurious Associations* (CISA), if there are unobserved spurious factors of variation $Z$, unobserved confounders $U$, and only $U$ is directly caused by domain $E$, so that $P_e(X, Y) = \int P_0(X, Y, Z|U)P_e(U)dZdU, \forall e \in \mathcal{E}$, where $P_0$ is a fixed distribution determined by the underlying dynamics of the system and $P_e(U)$ is a domain-specific distribution of the unobserved confounder.

Assumptions in domain shifts should be about the real-world processes that govern the domains from which the data is collected. This allows us to assess whether an assumption is reasonable for a given real-world situation. CISA is such an assumption; the spurious factors $Z$ and unobserved confounders $U$ are unknown and unobserved, but are assumed to correspond to real-world properties. This assumption about the real-world structure implies an assumption about the distributions $P_e(X, Y)$ in each domain. However, in general, the variables $Z$ and $U$ need not be identifiable from (i.e., uniquely determined by) the observational data. This point is only conceptually important, and does not affect the technical development in the following.

**CF-invariant representation learning** We now return to the question of what the right notion of domain-invariant representation is. In the case of CISA domains, there is a canonical notion for the part of $X$ that has a domain-invariant relationship with $Y$. Namely, $X_z^\perp$, the part of $X$ that is not affected by the spurious factors of variation. Accordingly, the goal of domain-invariant representation learning in this context is to find a representation $\phi$ such that $\phi(X)$ is counterfactually-invariant to $Z$ (or, equivalently, $\phi(X)$ is $X_z^\perp$ measurable). Of course, we could satisfy this condition by simply throwing away all of the information in $X$ (e.g., $\phi(X) = 0$ everywhere). So, we further look for the counterfactually-invariant representation that preserves the most predictive power for the label $Y$. Let $\Phi_{\text{cf-inv}}(\mathcal{E})$ denote the set of CF-invariant representations for CISA domains $\mathcal{E}$. Then, we take our domain-invariant representation learning objective to be:

$$\min_{\phi:\mathcal{X}\to\mathcal{H}, w:\mathcal{H}\to\mathcal{Y}} E_{P_{\mathcal{E}_{\text{train}}}}\left[L(Y, (w \circ \phi)(X))\right]$$

$$\text{subject to } \phi \in \Phi_{\text{cf-inv}}(\mathcal{E})$$

---

[3]Such a variable exists under weak conditions; e.g., $Z$ discrete (Veitch et al., 2021).

Here, the predictor $w$ and loss function $L$ capture the sense in which $\phi(X)$ should be predictive of $Y$.

The challenge here is that the spurious factors of variation are unknown and unobserved, so identifying the set of counterfactually-invariant representations is difficult. Now we can turn to understanding various approaches to domain adaptation methods as achievable relaxations of this ideal objective.

## 4 CAUSAL VIEW ON DOMAIN INVARIANT REPRESENTATION LEARNING

### 4.1 DATA AUGMENTATION

Data augmentation is a standard technique in machine learning pipelines, and has been shown to (sometimes) help when faced with domain shifts (Wiles et al., 2021). Our goal now is to understand when and why data augmentation might enable domain-invariant representation learning. The basic technique first applies pre-determined "label-preserving" transformations $t$ to original features $X$ to generate artificial data $t(X)$. There are two ways this transformed data can be used. The first option is to simply add the transformed data as extra data to a standard learning procedure. Alternatively, we might pass in pairs $(X_i, t(X_i))$ to our learning procedure, and directly enforce some condition that $\phi(X) \approx \phi(t(X))$ (Garg et al., 2019; Von Kügelgen et al., 2021). In both case, the natural questions are: What transformations are "label-preserving"? And, when do these techniques help with domain generalization?

We first formalize a notion of "label-preserving" for CISA domains. The key idea is that we can think of transformation $t(X)$ of $X$ as being equivalent to changing some cause of $X$ and then propagating this change through. For example, suppose a particular transformation $t$ rotates the input images by 30 degrees, and $W$ is the factor of variation corresponding to the angle away from vertical. Then, we can understand the action of $t$ as $t(X(w)) = X(w + 30)$, where we again use the potential outcomes notation for counterfactuals. With this idea in hand, we see that a transformation is *label-preserving* in CISA domains if it is equivalent to a change that affects only spurious factors of variation. That is, label-preserving transformations cannot affect $X_z^\perp$. Otherwise, the transformation may change the invariant relationship with $Y$; replacing the background of cows with "sand" with "grass" doesn't change animal type; but distorting the part corresponding to the "cow" object may.

**Definition 6.** We say a data transformation $t : \mathcal{X} \to \mathcal{X}$ is *label-preserving* for CISA domains $\mathcal{E}$ if, for each $X(z)$ there is $z'$ so that $t(X(z)) = X(z'), a.e..$

Label preserving transformations leave the CISA invariant relationships (between $X_z^\perp$ and $Y$) alone, but can change the relationship between $Y$ and the spurious factors of variation $Z$. Intuitively, if we have a 'large enough' set of such transformations, they can destroy the relationship between $Y$ and $Z$ that exists in the training data. So, we might expect that training with such data augmentation will automatically yield an optimal counterfactually-invariant predictor.

This is nearly correct, with the caveat that things can go wrong (for the naive training approach) if there is a part of $X$ causally related to both $Z$ and $Y$. That is, if there is a part of $X$ that relies on the interaction between $Z$ and $Y$. We follow Veitch et al. (2021) in formalizing how to rule out this case:

**Definition 7.** The spurious factors of variation $Z$ are *purely spurious* if $Y \perp\!\!\!\perp X | X_z^\perp, Z$

That is, without the unstable correlation (removed by conditioning on $Z$), $X_z^\perp$ is sufficient for $Y$. We can now state the main result connecting data augmentation and domain-invariance:

**Theorem 8.** *For a CISA domain, if the set of transformations $\mathcal{T}$ satisfies label-preserving and enumerates all potential outcomes of Z, then*

1. *If the model is trained to minimize risk on augmented data, and Z is purely spurious, or*
2. *If the model is trained to minimize risk on original data, with hard consistency regularization (i.e. enforcing $\phi(X) = \phi(t(X)), \forall t \in \mathcal{T}$)*

*Then we recover the CF-invariant predictor that minimizes risk on original data.*

Thus for CISA domains, ideal data augmentation (i.e. enumerating label-preserving transformations) will exactly learn CF-invariant representations. Moreover, this holds irrespective of what the true underlying causal structure is. Accordingly, such data augmentation would be the gold standard for domain-invariant representation learning.

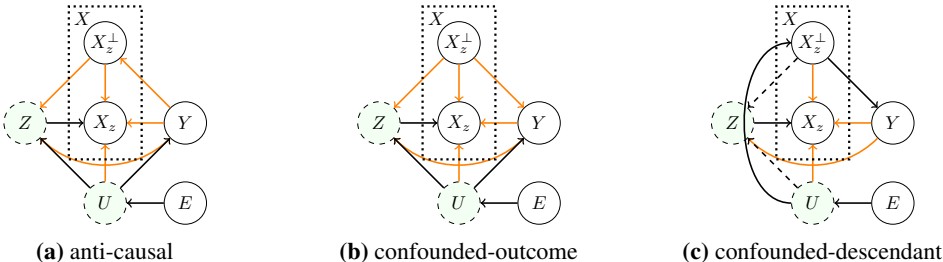

**Figure 2:** Every CISA compatible set of domains obeys one of the causal DAGs defined as follows: The DAG must match one of the three templates, where the black edges must be included, and the non-edges must be excluded. Orange edges may be included or excluded. In the case of Figure 2c, at least one of the two dashed black arrows must be included. (Note that in Figure 2a and Figure 2b, the edges between $X_z^\perp$ and $Y$ are typically included, as otherwise there is no part of $X$ that has a non-trivial and stable relalationship with $Y$.)

However, in practice, it can be hard to satisfy the idealized conditions. Applying predefined transformations without thinking about the specific applications can lead to violations of the label preserving condition. For example in a bird classification task, changing color may really change the bird species (this is called manifold intrusion in Guo et al. (2019)). Further, heuristic transformations often cannot enumerate all potential outcomes of $X(z)$. Indeed, deep models can sometimes memorize predetermined transformations and fail to generalize with previously unseen transformations (Vasiljevic et al., 2016; Geirhos et al., 2018).

Considering the limitations of using handcrafted transformations, a natural idea is to replace them with transformations learned from data. However, in practice, $Z$ is unknown and we only observe the data domains $E$. Then, learning transformations must rely either on detailed structural knowledge of the problem (Robey et al., 2021, e.g.,), or on some distributional relationship between $E$, $Y$ and $X$ and $t(X)$ (e.g., Goel et al., 2020). Since $t(X)$ is only used for the representation learning, this is equivalent to learning based on some distributional criteria involving $E$, $Y$, and $\phi(X)$—the subject of the next section.

## 4.2 DISTRIBUTIONALLY INVARIANT LEARNING

Many domain-invariant representation learning methods work by enforcing some form of distributional invariance Section 1. There are three possible notions of distributional invariance, each of which appears in the literature. Early work tries to learn a representation $\phi$ so that $\phi(X) \perp\!\!\!\perp E$ while $\phi(X)$ is still predictive of $Y$ (Muandet et al., 2013; Ganin et al., 2016). The intuition is that predictions must be based on features that cannot discriminate between training and test domains. Later work recognized that that $P(Y|X)$ may be unstable across domains, and instead aims at $\phi(X) \perp\!\!\!\perp E|Y$ (Li et al., 2018b; Long et al., 2018). Finally, some causally-motivated approaches have considered learning representations so that $Y \perp\!\!\!\perp E|\phi(X)$ (Peters et al., 2016; Wald et al., 2021).

When the representation $\phi(X)$ is predictive of $Y$ and $Y$ is dependent with $E$, these three independence statements—$\phi(X) \perp\!\!\!\perp E, \phi(X) \perp\!\!\!\perp E|Y, Y \perp\!\!\!\perp E|\phi(X)$—are the only possible distributional invariance relationships, which are usually mutually exclusive. The question now is: when, if ever, are each of these distributional invariances the right approach for domain-invariant representation learning?

**CISA compatible causal structures**    To answer this, the first step is to characterize the set of causal structures that are compatible with CISA.

**Theorem 9.** *Suppose a set of domains $\mathcal{E}$ share the common causal structure underlying $P_0(X, Y, Z|U)$. Then $\mathcal{E}$ satisfies CISA if and only if the corresponding causal DAG is one of the set given in Figure 2. In particular, there are three families of allowed DAGs: anti-causal, confounded-outcome, and confounded-descendant.*

The idea here is that for a set of CISA domains, the causal structure is the same in each domain. There are only three kinds of causal structure, distinguished by whether $Y$ causes $X_z^\perp$ or $X_z^\perp$ causes $Y$ (anti-causal vs confounded-outcome/descendant), and by whether the confounding affects $Y$ directly or just affects some (potential) causal descendant of $Y$ (confounded-outcome vs confounded-descendant).

**Anti-causal**  Image classification can be naturally viewed as an anti-causal problem. Various factors of variations such as lighting, background, angles, etc, and the object class $Y$, generate the image $X$. Some of the factors of variation $Z$ are confounded with $Y$—e.g., background and $Y$ may be associated due to evolutionary pressures. The Cow/Camel on Grass/Sand example fits here.[4]

**Confounded outcome**  The goal is to predict the helpfulness of a review. Each review receives a number of "helpful" votes $Y$, produced by site users. We use the review's text content $X$ as covariates. The data is collected for different types of products $E$. The model's performance drops significantly when deployed in new product type. We think that the general sentiment $Z$ of the review, and the helpfulness has unstable relationship across $E$: e.g. for books, customers write very positive reviews which are often voted favorably; for electronics this relationship is reversed.

**Confounded descendant**  Consider predicting unemployment rate $Y$ from a variety of economic factors $X$. It's not clear which factors cause $Y$ directly, and which are descendants of $Y$. The relationships among $X, Y$ change under certain big events, say financial crisis or pandemics. We denote those events as domains $E$, and take $U = E$. $U$ might affect $Y$'s descendants jointly with $Y$ (through intermediate variable $Z$), but $Y$ is not changed. Or $U$ might affect $\mathrm{pa}(Y)$ directly, which changes $Y$. Notice that in this case, the CF-invariant representation is also the representation that uses only $\mathrm{pa}(Y)$. Thus, for this causal structure, the counterfactually invariant notion matches the traditional causally-invariant representation learning desiderata (Peters et al., 2016; Arjovsky et al., 2019).

**Distributionally invariant learning**  We now return to the question of when distributional invariance (partially) enforces counterfactually-invariant representations. It turns out that the answer relies critically on the true causal structure of the problem:

**Theorem 10.** *If $\phi$ is a counterfactually-invariant representation,*

1. *if the underlying causal graph is anti-causal, $\phi(X) \perp\!\!\!\perp E | Y$;*
2. *if the underlying causal graph is confounded-outcome, $\phi(X) \perp\!\!\!\perp E$;*
3. *if the underlying causal graph is confounded-descendant, $Y \perp\!\!\!\perp E | \phi(X)$.*

*Remark* 11. This theorem looks similar to Veitch et al. (2021, Thm. 3.2). This is deceptive; here we observe the environment $E$, whereas they assumed observations of the spurious factors $Z$.

In words: each of the distributional invariances arises as a particular implication of CF-invariance. This suggests a learning strategy that relaxes CF-invariance. Namely, for a given problem, determine the distributional invariance implied by CF-invariance and then enforce that distributional invariance. That is, we learn a representation according to:

$$\min_{\phi \in \Phi_{\mathrm{DI}}(\mathcal{E}_{\mathrm{train}}), w} E_{P_{\mathcal{E}_{\mathrm{train}}}}[L(Y, (w \circ \phi)(X))],$$

where $\Phi_{\mathrm{DI}}(\mathcal{E})$ is the set of representations matching the causal structure of the domains.

Notice that the right distributional invariance is necessary but not sufficient for CF-invariance. There are two reasons: first, the causal condition of CF-invariance is not implied by its distributional signature in Theorem 10. Second, the distributional invariance can only be measured on a limited number of training domains, which may provide a limited constraints. In this sense, distributional invariance is a relaxation of CF-invariance; i.e., $\Phi_{\mathrm{cf\text{-}inv}}(\mathcal{E}) \subsetneq \Phi_{\mathrm{DI}}(\mathcal{E}_{\mathrm{train}})$.

Also notice that in general enforcing the wrong distributional invariance will directly contradict CF-invariance, and *increase* dependency on the spurious factors of variation. Distributional invariance is only a relaxation when we actually get the underlying causal structure correct!

### 4.3  INVARIANT RISK MINIMIZATION

The Invariant Risk Minimization (IRM) paradigm (Arjovsky et al., 2019) aims to find representations that admit a single predictor that has the optimal risk across all domains. That is, the set of IRM

---

[4]There is some controversy here since the label $Y$ is often due to human annotators (Lu et al., 2021).

representation is

$$\Phi_{\text{IRM}}(\mathcal{E}) := \{\phi : \exists\, w \text{ st } w \in \underset{\bar{w}}{\operatorname{argmin}}\, E_{P_e}[L(Y, (\bar{w} \circ \phi)(X))]\, \forall e \in \mathcal{E}\} \tag{4.1}$$

Here, the question is: when, if ever, does the IRM procedure yield a CF-invariant predictor?

Again the answer will turn out to depend on the underlying causal structure. IRM is justified in the case where $X$ includes both causes and descendants of $Y$, and the invariant predictor should use only information in the parents of $Y$. As explained above, this coincides with CF-invariance—and with distributional invariance—in the confounded-descendant case. Indeed, we can view IRM as a relaxation of $Y \perp\!\!\!\perp E|\phi(X)$. Instead of asking the distribution $P^e(Y|\phi(X) = h)$ to be invariant across domains for every $h \in \mathcal{H}$, we only require the risk minimizer under $P^e(Y|\phi(X) = h)$ to be invariant. So, a partial answer is that IRM is a relaxation of distributional invariance (and CF-invariance) when the underlying causal structure is confounded descendant.

However, the situation is harsher in the case of other causal structures. For anti-causal and confounded-outcome problems, the typical case is $\Phi_{\text{IRM}}(\mathcal{E}) = \emptyset$. For the anti-causal case, this is a consequence of *prior shift* (Zhang et al., 2013). We can change the optimal predictor just by changing the prior distribution on $Y$ without affecting the causal structure. Thus, there is no $X_z^\perp$-measurable predictor with invariant risk. The confounded-outcome case is even harder. In general, the generating function for $Y$ could involve interaction between $U$ and $X$. Then $P(Y|X_z^\perp)$ can be quite different because of the shifting $U$. Thus the risk minimizers may be very different across domains.

Fortunately, for anti-causal problems there is a natural generalization of IRM that partially enforces the distributional invariance. Similarly, we can relax the invariance of $P^e(\phi(X)|Y = y)$ to the invariance of a re-weighted risk. We define generalized-IRM (g-IRM) as follows, which is essentially importance sampling based on $Y$ (this simple procedure has causal implications (section 5)):

**Definition 12.** For anti-causal domains, we define the set of representations satisfying g-IRM as:

$$\Phi_{\text{g-IRM}}(\mathcal{E}, P_0) := \{\phi : \exists\, w \text{ st } w \in \underset{\bar{w}}{\operatorname{argmin}}\, E_{P_e}[\frac{P_0(Y)}{P_e(Y)} L(Y, (\bar{w} \circ \phi)(X))], \forall e \in \mathcal{E}\}$$

where $P_0(.)$ is a chosen baseline distribution for $Y$.

If we know the label marginal in the test domain, we could use that as $P_0$. Then the optimal predictor (on top of $\phi$) on training domains is optimal on the test domain. Next we can show the (generalized) IRM is a relaxation of the corresponding distributional invariance:

**Theorem 13.** *Let $\mathcal{E}$ satisfy CISA, then*

1. *if $\mathcal{E}$ is confounded-descendant, then $\Phi_{DI}(\mathcal{E}) \subset \Phi_{IRM}(\mathcal{E})$*
2. *if $\mathcal{E}$ is anti-causal, then $\Phi_{DI}(\mathcal{E}) \subset \Phi_{g\text{-}IRM}(\mathcal{E}, P_0)$ for any chosen $P_0$*

Although this simple generalization works for the anti-causal case, there is no such easy fix for the confounded-outcome case. For confounded outcome, the idea of having the same risk minimizer across domains does not make sense without further assumptions. The general case has $Y \leftarrow f(X, U, \eta)$ where $\eta$ is noise independent of $X, U$. For example when $U = E$ we can write $Y \leftarrow f_E(X, \eta)$ so there could be arbitrarily different relationships between $X$ and $Y$ in every domain. It may be possible to circumvent this by making structural assumptions on the form of $f$; e.g., there is an invariant risk minimizer in the case where the effect of $U$ and $X$ is additive (Veitch et al., 2021).

## 4.4 RELATIONSHIP OF METHODS

**Data augmentation training is the gold standard for CF-invariant representation learning** if it's possible to enumerate all label-preserving transformations (Theorem 8). This is impossible in general as it requires direct manipulation of the spurious factors $Z$. Still, using augmentation with label-preserving transformations (but not exhaustively) enforces a relaxation of CF-invariance.

**Distributional invariance** relaxes CF-invariance if and only if chosen to match the underlying causal structures (Theorem 10). This can be a good option when full augmentation is not possible.

**(generalized) IRM further relaxes distributional invariance** for anti-causal and confounded-descendant problem, when it's chosen to match the causal structure of the problem (Theorem 13). It

weakens the full independence criteria to use just the implication for a single natural test statistic: the loss of the model. This allows for more efficient algorithms (Arjovsky et al., 2019).

## 5 INSIGHTS FOR ROBUST PREDICTION

Often, learning domain-invariant representations is an intermediate step towards learning robust predictors. Here, robust means that the predictor should have good performance when deployed in a previously unseen domain. We now discuss the implications of our domain-invariant representation learning results for robust prediction.

**whether robust prediction is even possible depends on the underlying causal structure**  For confounded-descendant problems, robust prediction is straightforward. Enforcing the correct distributional invariance on representation function $\phi$ leads to invariant risk in test domains for every value of $\phi(X)$. If we enforce the weaker IRM requirement, the optimal predictor (on top of $\phi$) on training domains is optimal on test data. For anti-causal problems, there is no invariant risk minimizer in general because $P_e(Y)$ can change across domains — indeed, even in the most simple anti-causal problem in Appendix A.1, IRM fails. However, we can generalize IRM to match the causal structure: adjusting for prior shift using importance sampling, gIRM successfully recovers the optimal invariant predictor. For the confounded-outcome case, there is no notion of robust predictor without making some further structural assumptions.

**no method works for all domain generalization problems**  Heuristic data augmentation enforces CF-invariance directly regardless of causal structures, but requires truly label-preserving transformations and only solves the invariant representation problem if the transformation set affects all spurious factors. Distributionally-invariant methods can work, but each approach is only valid if it matches the underlying causal structure of the problem. Enforcing the wrong distributional invariance can actually harm performance (Veitch et al., 2021). Sometimes even seemingly innocent technique, such as importance sampling, has causal implications and could destroy invariant relalalionship (appendix A.2). Similarly, (generalized) invariant risk minimizer can work for some types of problems, but only when it matches the causal structure. This is consistent with the findings from various benchmark studies that there is no single method that does well in all tasks (Wiles et al., 2021; Koh et al., 2021; Gulrajani & Lopez-Paz, 2020).

**data augmentation helps in most cases**  Label-preserving data augmentation won't hurt domain generalization and can often help. This is true no matter the underlying causal structure of the problem. This matches empirical benchmarks where data augmentations usually help domain generalization performance, sometimes dramatically (Wiles et al., 2021; Koh et al., 2021; Gulrajani & Lopez-Paz, 2020). For example, Wiles et al. (2021) finds that simple augmentations used in Krizhevsky et al. (2012) generally improves performance when "augmentations approximate the true underlying generative model".

**pick a method matching the true causal structure**  Many papers apply distributional invariance approaches with no regard to the underlying causal structure of the problem. In particular, many tasks in benchmarks have the anti-causal structures, but the methods evaluated do not include those enforcing $\phi(X) \perp\!\!\!\perp E|Y$ (Koh et al., 2021). Tachet des Combes et al. (2020) find that methods enforcing $\phi(X) \perp\!\!\!\perp E|Y$ consistently improve over methods that enforce $\phi(X) \perp\!\!\!\perp E$—retrospectively, this is because they benchmark on anti-causal problems. Wiles et al. (2021) finds that learned data augmentation (Goel et al., 2020) consistently improves performance in deployment—this method can be viewed as enforcing $\phi(X) \perp\!\!\!\perp E|Y$; again, the benchmarks mostly have anti-causal structure.

## 6 CONCLUSION

We have introduced the CISA notion of domain shift and used it to study invariant representation learning methods. The key property of CISA is that it has a cannonical notion of "domain-invariant" representation: CF-invariance. We have seen that of data augmentation, distributional invariance learning, and risk invariant learning can be (sometimes) understood as relaxations of CF-invariance. The validity of the methods—as well as the connection between domain-invariance and domain-robust learning—depend fundamentally on the underlying causal structure of the data.

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

# A   EXPERIMENTAL DEMONSTRATION

A key point in the above analysis is that the invariant representation learning method we use must match the underlying causal structure of the data. In Section 5, we discuss this point in the context of existing large-scale, real-world experiments. However, such demonstrations rely on out of domain performance of various methods, which depends on both the match to the true causal structure but also on lower level implementation issues—e.g., hyperparameter tuning, overfitting, or optimization. This can make it somewhat difficult to draw precise conclusions.

To demonstrate the role of causal structure clearly we now study it in a simple case: two bit environments from Kamath et al. (2021). [5] This is a toy setting mimicing the well-known colored MNIST example used to demonstrate Invariant Risk Minimization Arjovsky et al. (2019). We create two domain shifts. The first is anti-causal and the second is confounded-descendant. As predicted from Theorem 10, we find that IRM fails but gIRM works in the first case, and vice versa in the second. [6]

In short, we find that even in the simplest possible case, causal structure plays a key role. In particular, vanilla IRM fails totally on an apparently innocuous modification of the data generating process (prior shift), and this is readily fixed by modifying the method to match the correct causal structure. Further, the experiments demonstrate that importance sampling based on $Y$—an apparently innocent technique—actually has causal implications and can destroy invariant relaltionship under certain causal structures (e.g. confounded-descendant in the second example).

## A.1   TWO-BIT-ENVS (ANTI-CAUSAL)

For each domain $e \in \mathcal{E}$, the data generating process is as follows:

$$Y \leftarrow \mathrm{Rad}(\gamma_e)$$
$$X_1 \leftarrow Y \cdot \mathrm{Rad}(\alpha)$$
$$X_2 \leftarrow Y \cdot \mathrm{Rad}(\beta_e)$$

where $\mathrm{Rad}(\pi)$ is a random variable taking value $-1$ with probability $\pi$ and $+1$ with probability $1 - \pi$.

This is a simplification of the ColoredMNIST problem (Arjovsky et al. (2019)): we know $X_1$ (corresponds to the digit shape) and $Y$ have invariant relationship: $P(X_1 = Y) = 1 - \alpha$. The correlation between $X_2$ (corresponds to color) and $Y$ is spurious, as $P(X_2 = Y) = 1 - \beta_e$ that varies across domains. The label imbalance across domains is due to prior-shift: $P(Y = -1) = \gamma_e$. We observe 4 training domains and predict on 1 test domain. We use $\alpha = 0.25$, and set $\beta_e = 0.1, 0.2, 0.15, 0.05$ in the training domains respectively, so that using the spurious correlation could get better in-domain performance. However, in the test domain the spurious correlation is flipped: we use $\beta_e = 0.9$ so the out-of-domain performance would be very bad if $X_2$ is used. Finally, we use $\gamma_e = 0.9, 0.1, 0.7, 0.3$ in training domains to create prior-shift. In the test domain the label is balanced ($\gamma_e = 0.5$). These 4 domains constitute $\mathcal{E}$. The goal is to find a predictor $f \in \{\{+1, -1\}^2 \rightarrow \mathcal{R}\}$ (prediction is $\hat{y} := \mathrm{sign}(f(x))$ for data $(x, y)$). The optimal 75% test accuracy is obtained when $f$ satisfies $f(1, \cdot) > 0$ and $f(-1, \cdot) \le 0$.

The original two-bit-envs problem has $\gamma_e = 0.5$ across domains, and IRM obtains one optimal predictor. However, it fails in this modified problem with prior-shift (Table 1). To understand and fix its failure, we can find a causal interpretation that both explains the data and is CISA-compatible: introduce $U, Z$ and set $Z \leftarrow U; U \leftarrow E$ as shown in Figure 3a. Then this domain shift problem falls under anti-causal subtype, and a CF-invariant predictor should only rely on $X_1$. IRM does not enforce the right invariance and fails to remove spurious $X_2$ as a result. Instead, we should use gIRM to (partially) enforce CF-invariance. Indeed, gIRM successfully forces the model to discard $X_2$ and obtain the optimal test accuracy (Table 1). Note that in the original two-bit-envs problem without prior-shift, both $X_1 \leftarrow Y \cdot \mathrm{Rad}(\alpha)$ and $Y \leftarrow X_1 \cdot \mathrm{Rad}(\alpha)$ can explain the data. Therefore we we

---

[5]Code would be made available upon accepted as a conference paper.

[6]Our experiment uses IRMv1 (and analogously gIRMv1), which is shown to fail with some choices of $\alpha$ Kamath et al. (2021) because of the relaxation from IRM to IRMv1. We avoid those choices of $\alpha$ so that we can focus on the high-level question instead of being distracted by the fragility of IRMv1. Below we use IRM and IRMv1 (also gIRM and gIRMv1) exchangebly.

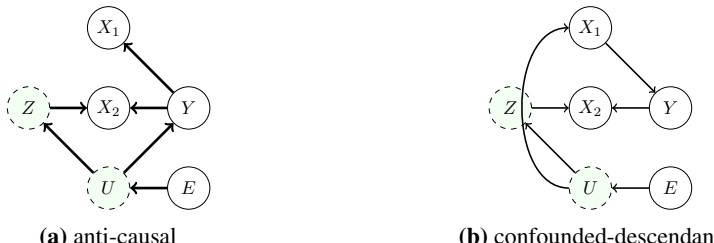

**(a)** anti-causal                **(b)** confounded-descendant

**Figure 3:** Causal graphs that are both are CISA-compatible, and can explain the data generating process (for Appendix A.1 and Appendix A.2 respectively). From the original data generating process, we introduced extra variables $U, Z$ and set $Z \leftarrow U$ and $U \leftarrow E$.

**Table 1:** In two-bit (anti-causal) experiment, IRMv1 predictor fails to discard spurious features because of prior-shift in $Y$. However, after modifying the method to match the underlying causal structure (using gIRMv1), we can recovers a CF-invariant predictor that obtains optimal test accuracy. The results are under cross-entropy loss (similar for for squared loss).

|            | $f_{\text{IRMv1}}$ | | $f_{\text{gIRMv1}}$ | |
|------------|------------|------------|------------|------------|
|            | $X_2 = +1$ | $X_2 = -1$ | $X_2 = +1$ | $X_2 = -1$ |
| $X_1 = +1$ | 2.53       | $-0.93$    | 1.16       | 1.08       |
| $X_1 = -1$ | $-0.08$    | $-3.19$    | $-1.11$    | $-1.03$    |

can *interpret* the data generating process as either anti-causal or confounded-descendant. So both IRM and gIRM (partially) enforce CF-invariance — in fact $\Phi_{\text{IRM}}(\mathcal{E}) = \Phi_{\text{g-IRM}}(\mathcal{E})$ so the resulting predictor is the same.

## A.2   TWO-BIT-ENVS (CONF-DESC)

Our implementation of enforcing gIRM regularization [7] is equivalent to: first, perform importance sampling with weight $w_e((x_1, x_2), y) = \frac{P_0(y)}{P_e(y)}$; next, enforce IRM regularization. Similarly, it's a common practice to perform importance sampling based on label $Y$ when it's imbalanced. However, as we shall show in this example, importance sampling can remove invariant features under certain causal structures.

In this example, the data generating process is as follows: for each domain $e \in \mathcal{E}$,

$$X_1 \leftarrow \text{Rad}(\gamma_e)$$
$$Y \leftarrow X_1 \cdot \text{Rad}(\alpha)$$
$$X_2 \leftarrow Y \cdot \text{Rad}(\beta_e)$$

Compared to the previous example, we do not change $P(X_1 = Y)$ and $P(X_2 = Y)$. The only change is on how the label imbalance is created: through covariate shift in $X_1$. We use the same parameters for $\alpha, \beta_e, \gamma_e$.

---

[7]There are two ways to implement gIRM. The first way is apply importance sampling to the regularization term only; the second way is to apply it to both the loss term and regularization term. In these two examples, the two implementations give the same result. Thus to better illustrate our point on importance sampling, we study the second implementation.

**Table 2:** In two-bit (confounded-descendant) experiment, directly applying IRMv1 recovers a CF-invariant predictor that obtains optimal test accuracy. On the contrary, applying importance sampling can destroy the invariant relationship in data — as a result, gIRMv1 learns only the trivial invariant predictor. The results are under cross-entropy loss (similar for for squared loss).

|            | $f_{\text{IRMv1}}$ | | $f_{\text{gIRMv1}}$ | |
|------------|------------|------------|------------|------------|
|            | $X_2 = +1$ | $X_2 = -1$ | $X_2 = +1$ | $X_2 = -1$ |
| $X_1 = +1$ | 1.1        | 1.1        | 0          | 0          |
| $X_1 = -1$ | $-1.1$     | $-1.1$     | 0          | 0          |

Again $P_e(Y)$ is different across domains, but this time gIRM forces the model to always predict $0$ (Table 2)! To understand why, we find a CISA-compatible DAG that explains the data as shown in Figure 3b (similarly introduce $U, Z$ and set $Z \leftarrow U; U \leftarrow E$). This is confounded-descendant and IRM enforces the right invariance whereas gIRM enforces the wrong one.

To understand how the importance sampling destroys even the invariant relationship between $X_1$ and $Y$, we look at the target distribution after reweighting (call it $Q_e$). Since the weighting function is $w_e(x_1, y) = \frac{P_0(x_1)}{P_e(y)}$ and that $w_e(x_1, y) = \frac{Q_e(x_1, y)}{P_e(x_1, y)}$, we have $Q_e(x_1, y) = P_e(x_1, y)\frac{P_0(y)}{P_e(y)}$. Observe that the probability $Q_e(X_1 = Y) = g(\gamma_e)$ where the $[0, 1]$-supported function $g$ (treat $\alpha$ as a constant and assume $\alpha > 0.5$) satisfies the following:

1. $g(\gamma) = g(1 - \gamma)$ so $g$ is symmetric around $0.5$.
2. $g$ strictly increases on $[0, 0.5]$ and strictly decreases on $[0.5, 1]$
3. $g(0) = g(1) = 0.5$, and $g(0.5) = 1 - \alpha$, so $g$ decreases as $\gamma_e$ deviates from $0.5$

Thus $0.5 < Q_1(X_1 = Y) = Q_2(X_1 = Y) < Q_3(X_1 = Y) = Q_4(X_1 = Y) < 1 - \alpha$. Therefore, importance sampling not only weakens the relalationship between $X_1$ and $Y$, but also makese it unstable! As a result, enforcing IRM on the resampled distribution finds no non-trivial invariant predictors.

## B PROOFS

**Theorem 8.** *For a CISA domain, if the set of transformations $\mathcal{T}$ satisfies label-preserving and enumerates all potential outcomes of $Z$, then*

1. *If the model is trained to minimize risk on augmented data, and $Z$ is purely spurious, or*
2. *If the model is trained to minimize risk on original data, with hard consistency regularization (i.e. enforcing $\phi(X) = \phi(t(X)), \forall t \in \mathcal{T}$)*

*Then we recover the CF-invariant predictor that minimizes risk on original data.*

*Proof.* First, for the convenience of notation let's assume $X = X(z_0)$ a.e. for some $z_0 \in \mathcal{Z}$. Then by the label-preserving $\mathcal{T}$, we have: for each $t \in \mathcal{T}$ we have $t(X)(= t(X(z_0))) = X(z)$ for some $z \in \mathcal{Z}$.

Consider consistency training. Let $\Phi_c(\mathcal{T})$ denote the set of representation functions satisfying consistency requirement under transformation set $\mathcal{T}$, i.e. $\Phi_c(\mathcal{T}) := \{\phi : \phi(X) = \phi(t(X)) \text{ a.e. } \forall t \in \mathcal{T}\}$. If $\phi \in \Phi_c(\mathcal{T})$, then for any $z, z' \in \mathcal{Z}$, can find $t \in \mathcal{T}$ such that $X(z') = t(X(z))$ since $\mathcal{T}$ enumerates all potential outcomes of $Z$; therefore $\phi(X(z')) = \phi(t(X(z))) = \phi(X(z))$ a.e. by consistency requirement. Thus $\phi \in \Phi_{\text{cf-inv}}(\mathcal{E})$. On the other hand if $\phi \in \Phi_{\text{cf-inv}}(\mathcal{E})$, then for any $t \in \mathcal{T}$, we have $\phi(t(X)) = \phi(X(z)) = \phi(X)$ for some $z \in \mathcal{Z}$. Thus $\phi \in \Phi_c(\mathcal{T})$. Therefore $\Phi_c(\mathcal{T}) = \Phi_{\text{cf-inv}}(\mathcal{E})$. Therefore, training the model to minimize risk on original data, with hard consistency regularization is equivalent to CF-invariant representation learning, which recovers the optimal CF-invariant predictor on training distribution.

Consider ERM training on augmented data with purely-spurious $Z$. Let $P$ denote the original distribution, and $\tilde{P}$ denote the distribution after the augmentation. Let $T$ be the random variable for transformation operation. First, the generating process of the augmented data is: first sample $T \sim \tilde{P}_T(.)$; then sample $(X, Y)|T = t$ from the distribution of $(t(X), Y)$. Then we have:

$$\tilde{P}(X, Y) = \int P(t(X), Y)d\tilde{P}_T(t)$$

$$= \int P(X(z), Y)d\tilde{P}_Z(z)$$

$$= \int P(X(z), Y(z))d\tilde{P}_Z(z)$$

$$= \int P(X, Y|do(z))d\tilde{P}_Z(z)$$

by the label-preserving of $\mathcal{T}$, and the fact that $Y$ is not a descendant of $Z$.

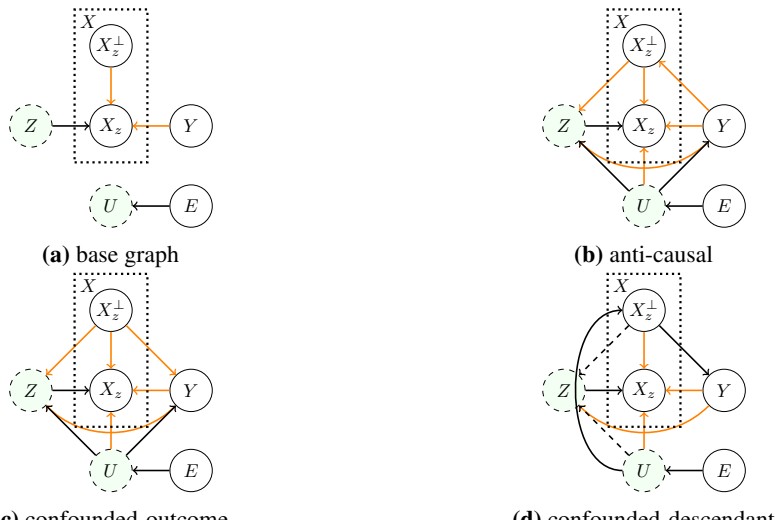

**(a)** base graph

**(b)** anti-causal

**(c)** confounded-outcome

**(d)** confounded-descendant

**Figure 4:** We put below Figure 2 again and the base graph for convenience of inspection. The black arrows are included in all graphs. The orange arrows are optional. At least one of the two dashed blue arrows in Figure 4d must exist.

Next, observe that $P(y|x, do(z)) = P(y|x_z^\perp)$. This is because: in original probability we have $Y \perp\!\!\!\perp X | X_z^\perp, Z$; the $do(z)$-operation removes the incoming edges of $Z$ and set $Z = z$; as a result $P(y|x, do(z)) = P(y|x_z^\perp, do(z)) = P(y|x_z^\perp)$.

Put together:

$$\tilde{P}(X, Y) = \int P(X, Y|do(z))d\tilde{P}(z)$$

$$= \int P(Y|X, do(z))P(X|do(z))d\tilde{P}(z)$$

$$= \int P(Y|X_z^\perp)P(X|do(z))d\tilde{P}(z)$$

$$= P(Y|X_z^\perp)\int P(X|do(z))d\tilde{P}(z) = P(Y|X_z^\perp)\tilde{P}(X)$$

Therefore the objective is:

$$E_{\tilde{P}}[L(Y, f(X))] = E_{\tilde{P}(X)}[E_{P(Y|X_z^\perp)}(L(Y, f(X)))]$$

Then for any input $x$, the the optimal predictor output $f^*(x) = \mathrm{argmin}_{a(x)} \int L(y, a(x))dP(y|x_z^\perp)$. This is the same as directly restricting predictor to be CF-invariant. □

**Theorem 9.** *Suppose a set of domains $\mathcal{E}$ share the common causal structure underlying $P_0(X, Y, Z|U)$. Then $\mathcal{E}$ satisfies CISA if and only if the corresponding causal DAG is one of the set given in Figure 2. In particular, there are three families of allowed DAGs: anti-causal, confounded-outcome, and confounded-descendant.*

*Proof.* There are a finite number of possible causal DAGs relating the variables $U, Z, X_z, Y, X_z^\perp$. Moreover, for a DAG to be compatible with CISA it must satisfy some conditions that narrows down the set. In particular, $Z$ causes $X_z$ but not $X_z^\perp$ or $Y$ (by definition of $Z$); $U$ should confound $Z$ and $Y$, but cannot confound $X_z^\perp$ and $Y$ (by definition of $U$); $E$ only affects $U$ (as only $P_e(U)$ changes across environments and $P(X, Y, Z|U)$ is invariant). Below we use these conditions to enumerate 5 all CISA-compatible DAGs.

Now we have $Z \to X_z$, $X_z^\perp \to X$ (optional) and $E \to U$ (and it's the only edge from $E$). Note that we define $Z$ to not have any causal effect on $Y$. Accordingly, the path $Z \to X_z \to Y$ is ruled out. Thus $X_z \to Y$ is not allowed but $Y \to X_z$ is optional. These edges form the base graph 4a to build upon.

**Figure 5:** Enumerating CISA-compatible causal graphs

Next, we divide into two cases: $X_z^\perp \to Y$ and $X_z^\perp \leftarrow Y$ (when there is no edge between them, we can treat it as either case and the resulting graphs are the same).

When $X_z^\perp \leftarrow Y$: we require that $U$ confounds $Z, Y$, so we need $U \to Y$ (otherwise $U$ can't cause $Y$) and $U \to Z$ (otherwise $U$ can't confound the relationship between $U, Z$). We do not allow $U \to X_z^\perp$ as otherwise the relationship between $X_z^\perp$ and $Y$ is confounded. There are a few optional edges $U \to X_z, Y \to Z, X_z^\perp \to Z$, as they do not violate CISA assumptions. Other edges cannot be allowed as they will violate CISA assumptions. These constitute the anti-causal subtype as illustrated in 4b.

When $X_z^\perp \to Y$, we can again divide into two exclusive cases: $U \to Y$ and $U \to X_z^\perp$. Why? We need at least one of these two edges, as otherwise $U$ does not cause $Y$; the two edges cannot exist simultaneously as otherwise the relationship between $X_z^\perp$ and $Y$ is confounded.

So, when $X_z^\perp \to Y$ and $U \to Y$: we need $U \to Z$ as otherwise $U$ does not confound $Z, Y$. There are a few optional edges $U \to X_z, Y \to Z, X_z^\perp \to Z$, as they do not violate CISA assumptions. Other edges cannot be allowed as they will violate CISA assumptions. These constitute the confounded-outcome subtype as illustrated in 4c.

Next, when $X_z^\perp \to Y$ and $U \to X_z^\perp$: to let $U$ cause $Z$ and confound $Z, Y$, we need at least one of the two edges (or both) $X_z^\perp \to Z, U \to Z$. There are a few optional edges $U \to X_z, Y \to Z$, as they do not violate CISA assumptions. Other edges cannot be allowed as they will violate CISA assumptions. These constitute the confounded-descendant subtype as illustrated in 4d.

$\square$

**Theorem 10.** *If $\phi$ is a counterfactually-invariant representation,*

1. *if the underlying causal graph is anti-causal, $\phi(X) \perp\!\!\!\perp E|Y$;*
2. *if the underlying causal graph is confounded-outcome, $\phi(X) \perp\!\!\!\perp E$;*
3. *if the underlying causal graph is confounded-descendant, $Y \perp\!\!\!\perp E|\phi(X)$.*

*Proof.* Reading d-separation from the corresponding DAGs, we have $X_z^\perp \perp\!\!\!\perp E|Y$ for anti-causal problems; $X_z^\perp \perp\!\!\!\perp E$ for confounded-outcome problems; $Y \perp\!\!\!\perp E|X_z^\perp$ for confounded-descendant problems. Since $\phi$ is CF-invariant, that means $\phi(X)$ is $X_z^\perp$-measurable. Thus the claim follows. $\square$

**Theorem 13.** *Let $\mathcal{E}$ satisfy CISA, then*

    *1. if $\mathcal{E}$ is confounded-descendant, then $\Phi_{DI}(\mathcal{E}) \subset \Phi_{IRM}(\mathcal{E})$*
    *2. if $\mathcal{E}$ is anti-causal, then $\Phi_{DI}(\mathcal{E}) \subset \Phi_{g\text{-}IRM}(\mathcal{E}, P_0)$ for any chosen $P_0$*

*Proof.* Confounded-descendant case: let $\phi \in \Phi_{\text{DI}}(\mathcal{E})$, i.e. $Y \perp\!\!\!\perp E | \phi(X)$. To show the risk minimizer is the same, it suffices to show $P_e(Y|\phi(X))$ to be the same for all $e \in \mathcal{E}$. This is immediate from the distributional invariance.

Anti-causal case: if the representation $\phi \in \Phi_{\text{DI}}(\mathcal{E})$, i.e. $\phi(X) \perp\!\!\!\perp E | Y$,

$$E_{P_e}\left[\frac{P_0(Y)}{P_e(Y)} L(Y, (\bar{w} \circ \phi)(X))\right] = E_{Y \sim P_e}\left[\frac{P_0(Y)}{P_e(Y)}[E_{\phi(X) \sim P_e(.|Y)}(L(Y, (\bar{w} \circ \phi)(X))|Y)]\right]$$
$$= E_{Y \sim P_0}[E_{\phi(X) \sim P(.|Y)}(L(Y, (\bar{w} \circ \phi)(X))|Y)]$$

The second equality is because $\phi(X) \perp\!\!\!\perp E | Y$.

Thus the objective function is the same across domains, so the optimal $w$ is the same. Therefore $\phi \in \Phi_{\text{g-IRM}}(\mathcal{E})$         $\square$

