# OpenReview forum: "A Unified Causal View of Domain Invariant Representation Learning"
_ICLR.cc/2023/Conference — Submitted to ICLR 2023_

### Official Review · Reviewer_1f5D · 2022-10-21

**Confidence:** 3
**Correctness:** 3
**Technical Novelty And Significance:** 3
**Empirical Novelty And Significance:** 3
**Recommendation:** 6

**Clarity, Quality, Novelty And Reproducibility:**

Clarity:
- In introduction, \phi is not defined.
- My understanding of Definition 1 is that the verbs “causing” and “affecting” mean both direct or indirect cause. Is this right? This should be made more explicit in the definition.
- Also in Definition 1, the word “latent” appears in parenthesis, suggesting that a spurious factor of variation can be observed. However, I believe the examples of Figure 1 contains only latent spurious factors of variations. It might be worth emphasizing this point in an example or somewhere in the text.
- Definition 2: I believe X(z) and X(z’) are random variables (because U | z and U| z’ are random). So is \phi(X(z)) = \phi(X(z’)) mean equality in distribution?
- Definition 3: I don’t understand why this definition is necessary, given that it was already said earlier that “X_z^⊥ denotes the part of X not affected by the spurious factors Z, and X_z the remaining part affected by Z”. Can’t we just think of X as the concatenation of X_z^⊥ and X_z? What am I missing?
- Just below Theorem 8, the confounded-outcome and confounded-descendant cases are contrasted: “... whether the confounding affects Y directly or just affects some causal descendant of Y (confounded-outcome vs confounded descendant).” This does not corresponds to the graph of figure 2c when all optional orange edges are removed since in that case, U confounds no descendant of Y. Am I misunderstanding something?
- Theorem 9 is introduced by saying it answers the question “when distributional invariance learns counterfactually-invariant representations”. But the implication of the theorem appears to be in the opposite direction, i.e. counterfactually invariance representation implies some distributional invariance (depending on the underlying causal graph). This means we could find a representation that is not CF-invariant. This point is confirmed right after theorem 9. This could be fixed simply by changing the introduction of the theorem. Also, should we worry about how "tight" this relaxation is? How likely is it that we find a representation that are very far from being CF-invariant?

Suggestions to improve the presentation:
- A reminder of what is meant by X-measurable would be helpful for people like myself who do not manipulate these notions regularly.
- Definition 4: Could the assumption that U does not confound the relationship between X_z^⊥ and Y be expressed as a specific factorization of the conditional P_0(X,Y,Z|U)? If so I believe it would be helpful to the reader to mention it.
- Definition 6: Might be useful to provide examples graphs such that Z is purely spurious or not. For example, which graphs in figure 1 satisfy this assumption?

Novelty/Originality: See my disclaimer.

Minor:
- Is there a typo in the second paragraph of 4.2? The authors seem to contrast the case where Y and E are independent VS dependent, but they mention “dependent” twice.
- Bad usage of \citep VS \citet.

**Strength And Weaknesses:**

Disclaimer:

I am not an expert on this topic and didn't know about most of the papers cited. As a consequence, I couldn't confirmed the accuracy of the claims made about those papers and cannot really assess how novel this contribution is. My review addresses mainly points about readability, motivation and clarity.

Strengths:
- This paper is well structured and for the most part easy to follow. I enjoyed reading it.
- The general goal of understanding theoretically under which settings previously proposed methods are expected to work is very important and should be valuable to the ICLR community.

Weaknesses:
- My main concern with this work is the weak motivation for exploring this specific class of data generating processes, i.e. CISA. A "cow on the beach” example is provided in the introduction to motivate CISA, but I could not understand how this example satisfies this assumption. It might be useful to provide more explanations and/or a graphical model in the appendix to illustrate the point, given how important this assumption is to the contribution. Right now, the CISA assumption seems arbitrary (at least to a non-expert).
- I found the introduction a bit hard to follow, I believe it could be shorten quite a bit to leave space for motivating CISA.
- Some mathematical statements lack clarity. I expand on that later in my review.





**Summary Of The Paper:**

This paper proposes a framework unifying multiple approaches for out-of-distribution generalization, namely, (i) data-augmentation, (ii) enforcing distributional invariances and (iii) invariant risk minimization. The framework is centered around the assumption that both the training and testing domains are "Causally Invariant with Spurious Associations" (CISA), which basically postulates the existence of a specific kind of causal graphical model with latent variables in which the causal relationships are assumed to be stable from one environment to another and only the distribution of a confounder is allowed to change. Based on this assumption, the authors propose "counterfactually invariant representation learning" as an ideal but generally intractable procedure to learn an invariant representation. Then, it is showed how the three approaches mentioned above can be seen as either exactly solving this problem or a relaxation of it, as long as the ground-truth generating process satisfies some assumptions (anti-causal, confounded-outcome & confounded-descendant) specific to each method. One of the key takeaway is that one should choose a method matching the true causal structure.

**Summary Of The Review:**

The paper was an interesting read, and I believe gaining a better theoretical understanding of precisely when these methods are expected to yield an invariant representation could provide guidance to practitioners. However, mainly because of the weak motivation for the CISA assumption (which is really central to this work), but also because of the lack of clarity at times, I believe this paper is only marginally above the acceptance threshold. I'm open to reconsider my evaluation after discussing with the authors as well as the other reviewers.

---

> ### Author Response · Authors · 2022-11-10
> **Reply to reviewer**
>
> Thank you for thorough review and questions! We’re glad you found the paper easy to follow, and the main goal important and of interest to the community.
>
> **Motivations for the CISA assumptions**
>
> * We want to focus on the type of domain shift induced by spurious correlations. In such problems, there is a part of $X$ that has a stable relationship with $Y$, whereas other parts of $X$ could be correlated with $Y$ through unobserved confounders that shift across domains. The purpose of CISA is that it is a particularly simple formalization of this idea.
>
> * Camel-Cow example: this example can be best modeled by the first graph in Fig 1---the anti-causal case. Here, the image $X$ is caused by some factors of variations: including background $Z$, and whether there is a camel or cow (labeled $Y$). In this situation, $Z$ is confounded with $Y$ in that cows ordinarily appear with grass, and camels with sand. However, this confounding relationship can change---cows on islands will often be on beaches, hence occurring with sand at a high rate. This example satisfies CISA in that the causal relationship between $X$ and its parents ($Y$ and $Z$) is stable, and only the unobserved confounder affecting the $Y$-$Z$ relationship changes across environments (locations).
>
> **Smaller points**
> * “Affects in def 1” It means both direct and indirect cause. We have modified the def to clarify.
> * With respect to the use of “(latents)”. For simplicity of presentation, we have removed the parentheses and always take the spurious factors to be latent
> * “ $\phi(X(z)) = \phi(X(z^{'}))$ mean equality in distribution?”: The equality is almost everywhere (def 2 is “ $\phi(X(z)) = \phi(X(z^{'}))$, a.e.”) Formally, this is a stronger notion than distributional equality.
> * About def 3:  we use def 3 to formalize “the part of $X$ not affected by $Z$” in a rigorous way. By “$X$-measurable”, you can think of it as “being a function of $X$”.  For the sake of understanding, it’s easier to think of $X$ as a concatenation of $X_z^\perp$ and $X_z$. But in general $X_z^\perp$ and $X_z$ are probably some more complex function (at least not linear) of $X$. For example, $X$ is an image, and to get the background part $X_z$ we normally apply some complex neural network instead of reading off fixed pixels in the image.
> * About the naming of confounded-outcome and confounded-descendant: the first has a direct edge from $U$ to the outcome $Y$ so we call it confounded-outcome. In the latter case, $U$ does not cause $Y$ directly (but hard to give a pithy name based on that!). However, $U$ affects (directly/indirectly) $X_z$, potentially a descendant of $Y$ — and in the DAG in IRM paper, one famous example for this case, $X_z$ is a descendant of $Y$. That’s why we call it conf-descendant. We have modified the prose to make it clearer.
> * About distributional invariance as a relaxation for CF-invariance: we have modified the sentence to resolve confusion. We acknowledge there is a gap between distributional invariance and CF-invariance. However, this is a fundamental restriction of using observational data. It could be an interesting direction for future work to study this gap. Note though that using the wrong distributional invariance will *contradict* counterfactual invariance. So, the right choice may not work, but the wrong choice will certainly make things fail.
> * About whether assumptions in def 4 (def 5 in the new version) could be expressed as a specific factorization of the joint probability: No, we can’t conclude the assumptions in def 5 with one specific factorization (assuming this is the factorization induced by causal relationship, i.e. $p(x_1, … x_p) = \Pi_i p(x_i | \text{pa}(x_i)))$. As we show in theorem 9, there are many DAGs compatible with the CISA assumption. Each of these implies a particular, incompatible, factorization.
> * About def 6 (purely spurious): it means $X_z$ is d-separable from $Y$ when conditioned on $X_z^\perp$ and $Z$. For example , it’s satisfied if we remove $Y \rightarrow X_z$ in Fig 1 (a). But this assumption does not limit the usefulness of data augmentation: when it's not satisfied, we should use consistency regularization instead of just ERM (thm 8)

---

> > ### Comment · Reviewer_1f5D · 2022-11-22
> > **Response**
> >
> > I thank the authors for their response. I'd like to ask for more details regarding motivation of CISA.
> >
> > **Motivation for CISA**
> >
> > I disagree that CISA is a *simple* formalization of the intuitive idea laid out in the response. It is certainly a formalization but I don't find it simple. Why so many latent variables are necessary? (related to reviewer eUgk's point about merging Z and U) And why this specific connectivity? Is it the "simplest" data-generating process accommodating all three types of out-of-distribution generalization? I.e (i) data-augmentation, (ii) enforcing distributional invariances and (iii) invariant risk minimization?

---

> > > ### Author Response · Authors · 2022-11-23
> > > **reply to reviewer**
> > >
> > > About the motivation for CISA:
> > > * For the invariant part of $X$ to make sense, there has to be some factor of variation that it is invariant to. So, we need latent $Z$ that's a cause of $X$
> > > * For $Z$ to even be interesting, it needs to be associated with Y (otherwise even ERM would just ignore it and there'd be no domain shift problem)
> > > * But $Z$ should not causally affect $Y$, otherwise it wouldn't make sense to be invariant to $Z$
> > > * Then, it's natural that $Z$ and $Y$ are associated due to a common cause ($U$)
> > > * To get a domain shift, it's then natural to let the strength of association between $Y$ and $Z$ vary (every other relationship is causal)
> > > CISA is just the immediate formalization of 1-5
> > >
> > > About merging Z and U:
> > > * $Z$ and $U$ are conceptually distinct as described above. This point is particularly useful when we consider data-augmentation: we want the data transformation $t$ to change $X$ by changing the factor of variation $Z$. If we merge $Z$ and $U$, $t$ changes $Y$ as well (since $U$ causes $Y$) — this is not “label preserving” anymore!

---

> > > > ### Comment · Reviewer_1f5D · 2022-12-02
> > > > **Satisfied with authors' response**
> > > >
> > > > Got it. Thanks for the clarification. It might be worth emphasizing this in the paper.
> > > >
> > > > I am satisfied with the authors responses and I'm thus keeping my score.

---

### Official Review · Reviewer_2pqk · 2022-10-24

**Confidence:** 3
**Clarity, Quality, Novelty And Reproducibility:** The paper is well written, and the pr…
**Correctness:** 3
**Technical Novelty And Significance:** 3
**Empirical Novelty And Significance:** 3
**Recommendation:** 6

**Strength And Weaknesses:**


In general, the paper is well written. The contribution is interesting and clear. The proposed model is sound and may theoretical analysis have been provided.
My major concerns are as follows, can we have some experiments to demonstrate the claims of the paper. At least, we can have some simulation studies. How can we ensure that the causal graph in Figure 1 is accurate. Since these causal graph is important for the following analysis, we should better provide more evidences on its rationalities.

**Summary Of The Paper:**

This paper aims to provide a causal view of the domain invariant learning. To achieve this goal, the authors have analyzed three types of methods. For all these methods, the authors believe that the invariant parts are different. By taking from the causal view, the authors have presented detail analysis on how to build a causality enhanced model to understand the domain invariant concept. In  the experiment, the authors have conducted extensive experiments to demonstrate the effectiveness of the model.


**Summary Of The Review:**

See the above comments.

---

> ### Author Response · Authors · 2022-11-10
> **Reply to reviewer**
>
> Thank you for your review---we’re glad you found the paper well written, interesting, and clear.
>
> **Experiments**
>
> Please see our reply to reviewer 2---we include a simulation study in the appendix. We have also added some additional references in the main text to points that are experimentally demonstrated.
>
> **How to ensure the correctness of causal diagrams?**
>
> By necessity, the DAGs in Fig1 and Fig 2 represent non-trivial assumptions about the world---sometimes, they won’t hold. Assessing this requires knowledge of the particular problem under consideration. Ideally, a practitioner should draw a DAG based on their knowledge of their problem, and check if they fall under one of the CISA DAGs in Fig 2. But, ultimately, like all causal assumptions, this judgment must be made problem by problem based on domain knowledge. (Note that this is often reasonable; e.g., whether or not Y causes X or X causes Y is often easy to judge).

---

> ### Author Response · Authors · 2022-12-05
> **Reply to reviewer**
>
> Do you have further concerns we can clarify? If we have fully addressed your concerns, we hope that you'll consider updating your score.

---

### Official Review · Reviewer_eUgk · 2022-10-24

**Confidence:** 3
**Correctness:** 3
**Technical Novelty And Significance:** 3
**Empirical Novelty And Significance:** Not applicable
**Recommendation:** 5

**Clarity, Quality, Novelty And Reproducibility:**

The paper is well-written and easy to follow. Investigating domain-invariant representation learning from a unified causal view seems new and could be of interest to the community.

**Strength And Weaknesses:**

**Strength:**

+ The paper proposes to look into different approaches to domain-invariant representation learning from a unified perspective
+ The idea seems straightforward and makes sense

**Weaknesses:**

- I am wondering if CF-invariance will lead to the optimal solution in terms of performance in different tasks. This is because some paper (e.g., https://openreview.net/forum?id=12RoR2o32T) suggests that CF-invariance be only sub-optimal in various scenarios.
- The causal diagrams shown in Figures 1&2 seem not flexible enough to cover many cases, e.g.,
   - Why do we only have $X_z^{\perp} \rightarrow X_z$, not $X_z^{\perp} \leftarrow X_z$?
   - Why is there no hidden confounders between $X_z^{\perp}$ and $X_z$ that is affected by $U$ or $E$?
- Is it really necessary to introduce both $Z$ and $U$ to the causal diagrams? Is it possible to just combine $Z$ and $U$ and treat it as one hidden variable that is affected by $U$ or $E$?
- I am unsure if it makes sense to simply divide $X$ into $X_z^{\perp}$ and $X_z$ in the causal diagrams? It might be better to explicitly include $X$?
- There is no empirical demonstrations for the theoretical results.

**Summary Of The Paper:**

This paper proposes from a unified perspective to investigate different approaches to learning domain-invariant representations. They first formulate CF-invariant representations and CISA domains, and then show different domain-invariant representation learning methods can be placed under the same framework.

**Summary Of The Review:**

My main concern is if CF-invariance will lead to the optimal models in various scenarios? If not, under which conditions? This point seems unclear and not discussed in the paper, which might weaken the impact of this paper. Also, how flexible and general are the causal diagrams considered in the paper?

---

> ### Author Response · Authors · 2022-11-10
> **Reply to reviewer**
>
> Thank you for your review---we’re glad you found the paper easy to follow, and the core idea to be novel and interesting.
>
> **About the optimality about CF-invariance**
>
> The claim here is not that learning an invariant representation will generally lead to good domain-shift robust predictors. Indeed, our results are generally somewhat negative on this question! (See section 5) For example, we find that CF-invariant prediction only corresponds to risk minimization invariance in the confounded-descendant case. Rather, the point of CF-invariance in this paper is that it gives a natural formalization of what invariant representation learning methods are actually trying to do.
>
> **Flexibility of the CISA framework**
>
> * “Why not $X_z^\perp \leftarrow X_z$ ?”: This is not allowed as otherwise $Z$ would be the causal parent of $X_z^\perp$, violating the definition of  $X_z^\perp$.
> * “Why is there no hidden confounders between $X_z^\perp$ and $X_z$ that’s affected by $U$ or $E$?”: this is actually allowed! In fact, this is conf-descendant case in Fig 2.
> * “Is it really necessary to introduce both $Z$  and $U$ to the causal diagrams? Is it possible to just combine  $Z$ and $U$ and treat it as one hidden variable that is affected by $U$ or $E$?”: $Z$ and $U$ are conceptually distinct objects (see for example our response to reviewer 1f5D about the camel-cow example); also at least some CISA allowed DAGs do not permit collapsing these variables (some graphs in conf-descendant case).
> * To address the flexibility of our framework: we study CISA primarily because it’s a flexible case where there is a well defined notion of invariant representation. The point is to use it as a tool for studying invariant representation learning methods in a context where there’s an expectation for how they “should” behave. Beyond that, CISA does seem reasonable for many situations where a domain shift is induced by spurious correlations; we have given some practical examples for CISA-compatible domain shifts in sec 4.2, which cover many of the popular problems included in the cited benchmark studies. We agree that there are realistic real-world scenarios that are not covered by CISA. In some cases, this is because there is simply no invariant structure (though variation may be weak enough to still get good domain-generalization results in practice), and in some cases this is because there are totally distinct flavors of domain shift (such as subpopulation shift notion of DRO).
>
> **Empirical evaluations**
>
> * As for empirical results, we think large-scale evaluation on real data is beyond the scope of this paper. Our goal here is to provide a theoretical framework for domain shifts that allows the study of connections between invariant learning approaches. To achieve this, we study idealized versions of the existing domain-generalization methods, ignoring issues like optimization performance, overfitting, and implementation choices. However, in practice, performance on real-world data is often dominated by such lower-level implementation details: how the particular invariance is enforced, how the hyperparameter is chosen. In fact, handling these are open questions in the domain generalization community! For example, IRMv1---the most studied and arguably simplest invariant learning algorithm---often fails to discard spurious features even in toy settings that match its assumptions (see, e.g. [1])!
> * We have tried to circumvent the difficulty of fair comparison by connecting our results to existing large-scale evaluations of domain-generalization methods---see the discussion of these studies in section 5.
> * We also include a two-bits-envs problem in the appendix as a clean illustration for our theoretical results: the invariant representation learning methods must match the underlying causal structure; otherwise, they either fail to discard spurious correlations, or destroy invariant structures. Due to the limit of space we put them in the appendix, but we have modified the main text to make more reference to this experiment.
>
> [1] Kamath et al, Does Invariant Risk Minimization Capture Invariance?

---

> ### Author Response · Authors · 2022-12-05
> **Reply to reviewer**
>
> Do you have further concerns we can clarify? If we have fully addressed your concerns, we hope that you'll consider updating your score.

---

### Official Review · Reviewer_nMg7 · 2022-11-02

**Confidence:** 4
**Correctness:** 2
**Technical Novelty And Significance:** 3
**Empirical Novelty And Significance:** Not applicable
**Recommendation:** 5

**Clarity, Quality, Novelty And Reproducibility:**

### Clarity
While the paper is, apart from a few typos and minor grammar mistakes, written in correct English, the writing is often unclear. Sometimes, it is difficult to follow the train of thought and more details or a clearer spelling out of the argument and the reasons why one should believe it would substantially improve this aspect. In fact, I see the clarity of presentation as the main weakness of the current paper.

Note that the paper violates several formatting requirements (not using the ICLR style file, or a variant thereof), figure formatting looks strange, no bibliography in the main submission, references are not correctly formatted (\citet{} vs \citep{} etc), math symbols are used inconsistently (orthogonal and independence symbol used interchangeably) and several other minor details all of which add up to leave a slightly sloppy impression.

### Quality
The quality of the work, to the extent that this can be judged given the above point, is (sufficiently) high and perfectly adequate for publication in a top ML conference.

### Novelty
The work is somewhat novel in extending and further developing existing work, specifically the key reference of Veitch et al. (2021), in a novel context; the connections it draws between different methods and underlying causal structure is of general interest to the causality and representation learning/domain generalisation communities.

### Reproducibility
Since the paper does not present empirical work (besides a toy experiment in the appendix which I did not check in detail), this does not seem to apply.

**Strength And Weaknesses:**

### Strengths
- The paper is well-motivated and original in pursuing a unified perspective of the by-now large literature on supervised invariant representation learning. To the best of my knowledge, no such unified account exists thus far.
- The categorisation of existing methods into three families (DA, DI, IR) is helpful.
- The paper succeeds to some extent in proposing a common framework for analysing different methods. I would agree that causality is the right language to do so, and the considered CISA setting is non-trivial and reasonably flexible in containing different causal structures as special cases.
- Some of the presented (theoretical) results are novel and interesting.
- The paper makes some prescriptive statements that can be useful for practitioners aiming to learn invariant representation while having available knowledge to decide which causal structure may closest to the truth.

### Weaknesses
- The paper is a bit hard to follow in parts; the presentation of the work could be substantially improved (see below for more details).
- Especially for being a theoretical/conceptual paper, attempting to provide a unifying account of existing literature, it often stays a bit too vague: certain key concepts such as "causally related", "confounded", "sufficient for Y" etc are not formally defined and, despite being intuitively familiar to someone with a background in causal inference, leave some ambiguity. A rigorous description of the entire considered data-generating process within a causal modelling framework such as SCMs or potential outcomes would be a helpful addition. Without specifying additional assumptions (e.g., faithfulness---is this assumed?), talking about causality solely based on the graphs in Figs. 1 and 2 is insufficient.
- Related to the previous two points, several claims made in the normal text are not made sufficiently clear and/or are not sufficiently backed by evidence or formal argument to allow for proper evaluation of their correctness. To give some examples: Sec 4.2 mentions confounding, but it is not clear exactly which variable is supposed to confound the relationship between which others; in the context of the examples for "anti-causal" and "confounded outcome", the arrows $X^\perp_z\to Z$ and $Y \to X_z$ seem strange; in Defn 11, should this hold for all P_0, or really just "any" P_0 (no additional requirements?), and why is P_0 not an argument to the LHS? much of the discussion in 4.3 and 5 is also not fully convincing/hard to follow. [I'm not claiming that these are necessarily errors or incorrect claims, just that it is hard to assess based on the presentation]
- The paper misses several references which I believe should be added: In the context of classical, simpler assumptions for domain adaptation, for example [R1,R2,R3] come to mind; it would also be interesting to see a discussion of other recent work providing a causal perspective on learning with spurious features [R4]; in the context of the line of work on IR learning, [R5] could also be mentioned, which can be seen as a generalisation of Krueger et al. (VRex); in the context of 4.1, [R6] also discusses generative process and counterfactual view for data augmentation that aims to learn invariant ("content") features and discard spurious "style" features; the term "anticausal" was coined in [R7]

[R1] Shimodaira, H. (2000). Improving predictive inference under covariate shift by weighting the log-likelihood function. Journal of statistical planning and inference, 90(2), 227-244.

[R2] Quinonero-Candela, J., Sugiyama, M., Schwaighofer, A., & Lawrence, N. D. (Eds.). (2008). Dataset shift in machine learning. Mit Press.

[R3] Lipton, Z., Wang, Y. X., & Smola, A. (2018, July). Detecting and correcting for label shift with black box predictors. In International conference on machine learning (pp. 3122-3130). PMLR.

[R4] Makar, M., et al. (2022, May). Causally motivated shortcut removal using auxiliary labels. In International Conference on Artificial Intelligence and Statistics (pp. 739-766). PMLR.

[R5] Eastwood, C., et al. "Probable domain generalization via quantile risk minimization." arXiv preprint arXiv:2207.09944 (2022).

[R6] Von Kügelgen, J., et al. (2021). Self-supervised learning with data augmentations provably isolates content from style. Advances in neural information processing systems, 34, 16451-16467.

[R7] Schölkopf, B., et al. (2012, January). On causal and anticausal learning. In ICML.

### Comments
- The proofs of Thms. 9 and 12 seem rather trivial in that the result essentially follows from the previous assumptions and definitions; hence, it may be more appropriate to call these Lemma or Proposition.
- for the more involved proofs, it would be helpful to include at least a sketch in the main paper
- since representation learning is often associated with an unsupervised setting, I think it would be good to add "supervised" to the title (it is already stated in the introduction that this work only focuses on *supervised* domain invariant representation learning)
- the description of the "two-bit-envs experiment" seems out of place in the conclusion and lacks context and details

**Summary Of The Paper:**

The paper aims to present a unifying account of different methods for domain-invariant (supervised) representation learning, that is, given data from multiple joint distributions over inputs X and labels Y, learning structure that is stable across them and discarding "spurious" parts that vary. Specifically, three families of methods are considered: data augmentation (DA), distributional invariance (DI), and invariant risk (IR). To analyse these methods, the paper considers a setting where different domains or datasets are related through the CISA assumption, which postulates that causal relationships stay the same while only the distribution of a latent common cause U changes across domains. More formally, any $P_e(X,Y)$ is obtained from the same $P_0(X,Y,Z|U)$ via integration w.r.t.  a domain-varying $P_e(U)$ and marginalisation of the latent factors of variation (FoV) $Z$. The paper uses the notion of counterfactual invariance (CF-invariance) of Veitch et al. (2021) to formalise the desired notion of invariance. It then proceeds to study the implications of learning via DA, DI, and IR under different causal structures satisfying the CISA assumption.

It demonstrates that the causal structure is important in determining whether or not different methods (and subvariants thereof) succeed in learning domain invariant structure in the form of a CF-invariant representation of the input. Roughly, it is argued that DA is the gold standard but typically infeasible, depending on the causal structure, different types of DI are necessary (but not sufficient) for CF-invariance, and IR further relaxes DI by only enforcing invariance w.r.t. aspects of the distribution on which the risk depends. The paper concludes by discussing its results and insights in the context of robust prediction and related literature.

**Summary Of The Review:**

The paper is original and tackles an important problem from a new and interesting angle. I see great potential for this work. However, the current presentation is in many parts too vague and unclear and sometimes fails to convincingly deliver the main points---even if these may be correct. I view this submission as a borderline case. Since the paper could be greatly improved through careful revision, rewriting, and reformatting, I lean toward rejection.

---

> ### Author Response · Authors · 2022-11-10
> **Reply to reviewer**
>
> Thank you for the very detailed review and helpful comments. We’re glad that you find the paper well-motivated and original, and the CISA framework to be useful for studying invariant supervised representation learning.
>
> With respect to clarity, there seem to be three distinct things at play.
> * There are indeed some places in the paper where there are indeed unclear sentences or typos and formatting issues. We appreciate your very thorough reading! We have fixed these where you’ve pointed them; detailed below.
> * The definition of the CISA framework and the underlying causal assumptions. We have addressed this by clarifying the presentation of theorem 8 (theorem 9 in the new version). We discuss this below.
> * The use of relatively informal language in the prose of the paper. This seems to be a stylistic preference. We believe that the content of the formal definitions and theorem statements---the contributions of the paper---are precise and correct. The prose of the text is often somewhat informal with the aim of explaining the underlying intuitions without getting bogged down in technicalities. For example, the use of “confounder” the refer to the variable U that is used throughout the paper as the unobserved confounder, or the use of “causally related” to capture any causal relationship (e.g., $Y$ causes $X_z$) that leads to a violation of $Y \perp X | X_z^\perp, Z$. In all cases, the informal language comes in the prose around formal statements that make the relevant concepts precise.
>
> **Causal Modeling framework of DGP**
>
> With respect to “A rigorous description of the entire considered data-generating process within a causal modeling framework such as SCMs or potential outcomes would be a helpful addition. Without specifying additional assumptions (e.g., faithfulness---is this assumed?), talking about causality solely based on the graphs in Figs. 1 and 2 is insufficient."
>
> The key obstacle here is that CISA defines a *set* of causal structures (and associated DGPs). Namely, it's a formalization of the set of causal structures that is compatible with "domain shifts are due to spurious associations". This is naturally an implicit definition---we allow causal structures compatible with this assumption. It is not trivial to simply enumerate all the SCMs compatible with the assumption. Indeed, this enumeration is precisely the content of theorem 9! That is, the structural causal models allowed under the CISA assumption are exactly those with a corresponding causal DAG in the set identified by theorem 9. This is one of the key contributions of this paper: proposing an intuitive formalization for a class of domain shifts, and characterizing the SCMs that satisfies it.
>
> In other words, to check if a given SCM is compatible with CISA, one can draw the corresponding DAG, and check if it’s in the set identified by theorem 9.
>
> To clarify this, we have
> 1. Clarified the definition of CISA by adding a formal definition (def 4) for the unobserved confounder U, and
> 2. reworded theorem 9, the corresponding figure 2, and the surrounding prose to make this point more clearly.
>
> (We don’t require faithfulness because the only non-edges in theorem 9 are directly from the definition of CISA. All other results rely on the markov property of d-separation)
>
>
> **Further Questions**
>
> "Sec 4.2 mentions confounding, but it is not clear exactly which variable is supposed to confound the relationship between which others" . This confounding is mentioned immediately after theorem 8, which makes this precise (in figure 2)---the confounder is U in the causal DAGs. We have also updated the prose to clarify this point. See also the newly added definition 4.
>
> “In the context of the examples for "anti-causal" and "confounded outcome", the arrows $X_z^{\perp} \rightarrow Z$ and $Y \rightarrow X_z$ seem strange”. Indeed some cases might be unrealistic in the real world. But here we are just enumerating all allowed DAGs. The purpose for Thm 9 (thm 8 in the original version) is: given a SCM for a particular problem, we draw its DAG and see if it falls under one of the graphs in Fig 2. This may allow, in particular, some graphs that are not very useful for real applications.
>
> About the “two-bit-envs experiment”. We have removed the reference in the conclusion, and instead include references to the experiment in sections 4.3 and 5 where we make claims that are verified by the experiment. (Ideally, the experiment would live in the main paper, but it has been relegated to the appendix for space)
>
> About further confusions on sec 4.3 and sec 5, could you elaborate?
>
> As for “The proofs of Thms. 9 and 12 seem rather trivial … may be more appropriate to call these Lemma or Proposition”. Indeed the proofs are simple, but we view it as a feature not a bug. The results are conceptually important.

---

> ### Author Response · Authors · 2022-12-05
> **Reply to reviewer**
>
> Do you have further concerns we can clarify? If we have fully addressed your concerns, we hope that you'll consider updating your score.

---

### Author Response · Authors · 2022-11-18
**Thanks to AC and reviewers**

We thank all reviewers for their comments and questions. We think the paper has improved as a result!

All reviewers agree that this paper is well-motivated and the contribution is original and useful.

While reviewer 1 was concerned about clarity, the other reviewers found it “well-written”, “easy to follow” and “straightforward”. Reviewer 1 helpfully identified some particular unclear sentences and formatting issues, and we have modified the paper as suggested to make it clearer (see our reply to  Reviewer 1). Additionally, we have clarified the presentation of the CISA definition and Theorem 9 (formerly 8) to more clearly explain the underlying structural assumption of CISA. (In short: just giving such a characterization directly is non-trivial, and indeed this is what theorem 6 does!) We think the paper is improved, and we thank reviewer 1 again for their careful reading and comments!

Reviewer 2 was concerned about whether counterfactual invariance is actually optimal for handling domain shifts. In our response, we have clarified that we do not make this assertion—indeed, our results show that the connection between invariance and robustness is generally complex! Rather, counterfactual invariance formalizes a common intuition about invariant representation learning that is often relied upon. This formalization allows us to recognize the non-optimality precisely (as well as circumstances where invariance is indeed an appropriate way of achieving robustness).

---

### Decision · Program_Chairs · 2023-01-20

**Decision:**

Reject

**Justification For Why Not Higher Score:**

The presentation seems too vague and unclear, preventing reviewers from evaluating the value of the contribution precisely. Moreover, several closely related works in the literature, including the conditionally transferrable component-based method, are overlooked.

**Justification For Why Not Lower Score:**

Reviewers agree that the perspective may have a lot of potential.

**Metareview: Summary, Strengths And Weaknesses:**

This paper provides a unifying account of different methods for domain-invariant (supervised) representation learning. Reviewers agree that the perspective may have a lot of potential, but at the same time, the presentation seems too vague and unclear, preventing reviewers from evaluating the value of the contribution precisely. Moreover, several closely related works in the literature, including the conditionally transferrable component-based method, are overlooked.